# Implementation Matters in Deep Policy Gradients: A Case Study on PPO and TRPO

Logan Engstrom[*], Andrew Ilyas[*], Shibani Santurkar[1], Dimitris Tsipras[1],
Firdaus Janoos[2], Larry Rudolph[1,2], and Aleksander Mądry[1]

[1]MIT    [2]Two Sigma
{engstrom,ailyas,shibani,tsipras,madry}@mit.edu
rudolph@csail.mit.edu, firdaus.janoos@twosigma.com

## Abstract

We study the roots of algorithmic progress in deep policy gradient algorithms through a case study on two popular algorithms: Proximal Policy Optimization (PPO) and Trust Region Policy Optimization (TRPO). Specifically, we investigate the consequences of "code-level optimizations:" algorithm augmentations found only in implementations or described as auxiliary details to the core algorithm. Seemingly of secondary importance, such optimizations turn out to have a major impact on agent behavior. Our results show that they (a) are responsible for most of PPO's gain in cumulative reward over TRPO, and (b) fundamentally change how RL methods function. These insights show the difficulty, and importance, of attributing performance gains in deep reinforcement learning.

## 1 Introduction

Deep reinforcement learning (RL) algorithms have fueled many of the most publicized achievements in modern machine learning (Silver et al., 2017; OpenAI, 2018; Abbeel & Schulman, 2016; Mnih et al., 2013). However, despite these accomplishments, deep RL methods still are not nearly as reliable as their (deep) supervised learning counterparts. Indeed, recent research found the existing deep RL methods to be brittle (Henderson et al., 2017; Zhang et al., 2018), hard to reproduce (Henderson et al., 2017; Tucker et al., 2018), unreliable across runs (Henderson et al., 2017; 2018), and sometimes outperformed by simple baselines (Mania et al., 2018).

The prevalence of these issues points to a broader problem: we do not understand how the parts comprising deep RL algorithms impact agent training, either separately or as a whole. This unsatisfactory understanding suggests that we should re-evaluate the inner workings of our algorithms. Indeed, the overall question motivating our work is: how do the multitude of mechanisms used in deep RL training algorithms impact agent behavior?

**Our contributions.** We analyze the underpinnings of agent behavior—both through the traditional metric of cumulative reward, and by measuring more fine-grained algorithmic properties. As a first step towards tackling this question, we conduct a case study of two of the most popular deep policy-gradient methods: Trust Region Policy Optimization (TRPO) (Schulman et al., 2015a) and Proximal Policy Optimization (PPO) (Schulman et al., 2017). These two methods are closely related: PPO was originally developed as a refinement of TRPO.

We find that much of the PPO's observed improvement in performance comes from seemingly small modifications to the core algorithm that either can be found only in a paper's original implementation, or are described as auxiliary details and are *not* present in the corresponding TRPO baselines. [1] We pinpoint these modifications, and perform an ablation study demonstrating that they are instrumental to the PPO's performance.

---

[*]Equal contribution. Work done in part as an intern at Two Sigma.

[1]Note that these code-level optimizations are separate from "implementation choices" like the choice of PyTorch versus TensorFlow in that they intentionally change the training algorithm's operation.

This observation prompts us to study how such code-level optimizations change agent training dynamics, and whether we can truly think of them as merely auxiliary improvements. Our results indicate that these optimizations fundamentally change algorithms' operation, and go even beyond improvements in agent reward. We find that they majorly impact a key algorithmic principle behind TRPO and PPO's operations: trust region enforcement.

Ultimately, we discover that the *PPO code-optimizations are more important in terms of final reward achieved* than the choice of general training algorithm (TRPO vs. PPO). This result is in stark contrast to the previous view that the central PPO clipping method drives the gains seen in Schulman et al. (2017). In doing so, we demonstrate that the algorithmic changes imposed by such optimizations make rigorous comparisons of algorithms difficult. Without a rigorous understanding of the full impact of code-level optimizations, we cannot hope to gain any reliable insight from comparing algorithms on benchmark tasks.

Our results emphasize the importance of building RL methods in a modular manner. To progress towards more performant and reliable algorithms, we need to understand each component's impact on agent behavior and performance—both individually, and as part of a whole.

Code for all the results shown in this work is available at `https://github.com/MadryLab/implementation-matters`.

## 2 RELATED WORK

The idea of using gradient estimates to update neural network–based RL agents dates back at least to the work of Williams (1992), who proposed the REINFORCE algorithm. Later, Sutton et al. (1999) established a unifying framework that casts the previous algorithms as instances of the policy gradient method.

Our work focuses on proximal policy optimization (PPO) (Schulman et al., 2017) and trust region policy optimization (TRPO) (Schulman et al., 2015a), which are two of the most prominent policy gradient algorithms used in deep RL. Much of the original inspiration for the usage of the trust regions stems from the conservative policy update of Kakade (2001). This policy update, similarly to TRPO, uses a natural gradient descent-based greedy policy update. TRPO also bears similarity to the relative policy entropy search method of Peters et al. (2010), which constrains the distance between marginal action distributions (whereas TRPO constrains the conditionals of such action distributions).

Notably, Henderson et al. (2017) points out a number of brittleness, reproducibility, and experimental practice issues in deep RL algorithms. Importantly, we build on the observation of Henderson et al. (2017) that final reward for a given algorithm is greatly influenced depending on the code base used. Rajeswaran et al. (2017) and Mania et al. (2018) also demonstrate that on many of the benchmark tasks, the performance of PPO and TRPO can be matched by fairly elementary randomized search approaches. Additionally, Tucker et al. (2018) showed that one of the recently proposed extensions of the policy gradient framework, i.e., the usage of baseline functions that are also action-dependent (in addition to being state-dependent), might not lead to better policies after all.

## 3 ATTRIBUTING SUCCESS IN PROXIMAL POLICY OPTIMIZATION

Our overarching goal is to better understand the underpinnings of the behavior of deep policy gradient methods. We thus perform a careful study of two tightly linked algorithms: TRPO and PPO (recall that PPO is motivated as TRPO with a different trust region enforcement mechanism). To better understand these methods, we start by thoroughly investigating their implementations in practice. We find that in comparison to TRPO, the PPO implementation contains many non-trivial optimizations that are not (or only barely) described in its corresponding paper. Indeed, the standard implementation of PPO [2] contains the following additional optimizations:

---

[2]From the OpenAI baselines GitHub repository: `https://github.com/openai/baselines`

1. **Value function clipping:** Schulman et al. (2017) originally suggest fitting the value network via regression to target values:

$$L^V = (V_{\theta_t} - V_{targ})^2,$$

   but the standard implementation instead fits the value network with a PPO-like objective:

$$L^V = \min \left[ (V_{\theta_t} - V_{targ})^2, \left( \text{clip} \left( V_{\theta_t}, V_{\theta_{t-1}} - \varepsilon, V_{\theta_{t-1}} + \varepsilon \right) - V_{targ} \right)^2 \right],$$

   where $V_\theta$ is clipped around the previous value estimates (and $\varepsilon$ is fixed to the same value as the value used in (2) to clip the probability ratios).

2. **Reward scaling:** Rather than feeding the rewards directly from the environment into the objective, the PPO implementation performs a certain discount-based scaling scheme. In this scheme, the rewards are divided through by the standard deviation of a rolling discounted sum of the rewards (without subtracting and re-adding the mean)—see Algorithm 1 in Appendix A.2.

3. **Orthogonal initialization and layer scaling:** Instead of using the default weight initialization scheme for the policy and value networks, the implementation uses an orthogonal initialization scheme with scaling that varies from layer to layer.

4. **Adam learning rate annealing:** Depending on the task, the implementation sometimes anneals the learning rate of Adam (Kingma & Ba, 2014) (an already adaptive method) for optimization.

5. **Reward Clipping**: The implementation also clips the rewards within a preset range (usually $[-5, 5]$ or $[-10, 10]$).

6. **Observation Normalization**: In a similar manner to the rewards, the raw states are also not fed into the optimizer. Instead, the states are first normalized to mean-zero, variance-one vectors.

7. **Observation Clipping**: Analagously to rewards, the observations are also clipped within a range, usually $[-10, 10]$.

8. **Hyperbolic tan activations**: As also observed by Henderson et al. (2017), implementations of policy gradient algorithms also also use hyperbolic tangent function activations between layers in the policy and value networks.

9. **Global Gradient Clipping**: After computing the gradient with respect to the policy and the value networks, the implementation clips the gradients such the "global $\ell_2$ norm" (i.e. the norm of the concatenated gradients of all parameters) does not exceed $0.5$.

These optimizations may appear as merely surface-level or insignificant algorithmic changes to the core policy gradient method at hand. However, we find that they dramatically affect the performance of PPO. To demonstrate this, we start by performing a full ablation study on the four optimizations mentioned above [3]. Figure 1 shows a histogram of the final rewards of agents trained with every possible configuration of the above optimizations—for each configuration, a grid search for the optimal learning rate is performed, and we measure the reward of random agents trained using the identified learning rate. Our findings suggest that many code-level optimizations are necessary for PPO to attain its claimed performance.

The above findings show that our ability to understand PPO from an algorithmic perspective hinges on the ability to distill out its fundamental principles from such algorithm-independent (in the sense that these optimizations can be implemented for any policy gradient method) optimizations. We thus consider a variant of PPO called PPO-MINIMAL (PPO-M) which implements only the core of the algorithm. PPO-M uses the standard value network loss, no reward scaling, the default network initialization, and Adam with a fixed learning rate. Importantly, PPO-M ignores all the code-level optimizations listed above in the beginning of Section 3. We then explore PPO-M alongside PPO and TRPO. We list all the algorithms we study and their defining properties in Table 1.

Overall, our results on the importance of these optimizations both corroborate results demonstrating the brittleness of deep policy gradient methods, and demonstrate that even beyond environmental brittleness, the algorithms themselves exhibit high sensitivity to implementation choices [4].

---

[3]Due to restrictions on computational resources, we could only perform a full ablation on the first four of the identified optimizations.

[4]This might also explain the difference between different codebases observed in Henderson et al. (2017)

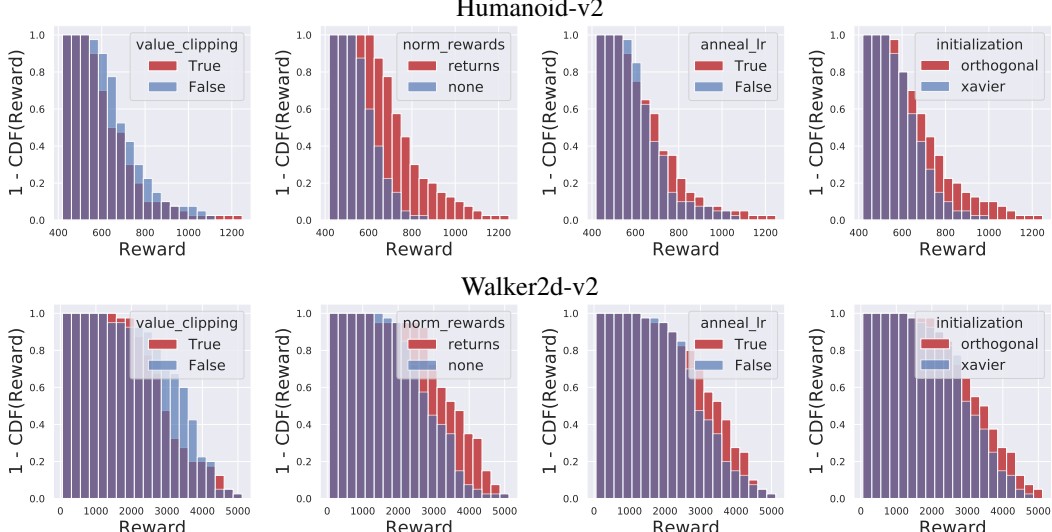

Figure 1: An ablation study on the first four optimizations described in Section 3 (value clipping, reward scaling, network initialization, and learning rate annealing). For each of the $2^4$ possible configurations of optimizations, we train a Humanoid-v2 (top) and Walker2d-v2 (bottom) agent using PPO with five random seeds and a grid of learning rates, and choose the learning rate which gives the best average reward (averaged over the random seeds). We then consider all rewards from the "best learning rate" runs (a total of $5 \times 2^4$ agents), and plot histograms in which agents are partitioned based on whether each optimization is on or off. Our results show that reward normalization, Adam annealing, and network initialization each significantly impact the rewards landscape with respect to hyperparameters, and were necessary for attaining the highest PPO reward within the tested hyperparameter grid. We detail our experimental setup in Appendix A.1.

Table 1: List of algorithms studied in this work, with their crucial properties. Step method refers to the method used to build each training step, PPO clipping refers to the use of clipping in the step (as in Equation (2)), and PPO optimizations refer to the optimizations listed in Section 3.

| Algorithm | Section | Step method | Uses PPO clipping? | Uses PPO optimizations? |
|---|---|---|---|---|
| PPO | — | PPO | ✓ | As in (Dhariwal et al., 2017) |
| PPO-M | Sec. 3 | PPO | ✓ | ✗ |
| PPO-NoClip | Sec. 4 | PPO | ✗ | Found via grid search |
| TRPO | — | TRPO | — | ✗ |
| TRPO+ | Sec. 5 | TRPO | — | Found via grid search |

## 4 CODE-LEVEL OPTIMIZATIONS HAVE ALGORITHMIC EFFECTS

In the previous section, we found that canonical implementations of PPO contain many *code-level optimizations*: implementation choices that are not integral to the method but profoundly impact performance.

The seemingly disproportionate effect of code-level optimizations identified in our ablation study may lead us to ask: *how do these seemingly superficial code-level optimizations impact underlying agent behavior?* In this section, we demonstrate that the code-level optimizations fundamentally alter agent behavior. Rather than merely improving ultimate cumulative award, such optimizations directly impact the principles motivating the core algorithms.

**Trust Region Optimization.** A key property of policy gradient algorithms is that update steps computed at any specific policy $\pi_{\theta_t}$ are only guaranteed predictiveness in a neighborhood around $\theta_t$. Thus, to ensure that the update steps we derive remain predictive many policy gradient algo-

rithms ensure that these steps stay in the vicinity of the current policy. The resulting "trust region" methods (Kakade, 2001; Schulman et al., 2015a; 2017) try to constrain the local variation of the parameters in policy-space by restricting the distributional distance between successive policies.

A popular method in this class is trust region policy optimization (TRPO) Schulman et al. (2015a). TRPO constrains the KL divergence between successive policies on the optimization trajectory, leading to the following problem:

$$\max_{\theta} \quad \mathbb{E}_{(s_t, a_t) \sim \pi} \left[ \frac{\pi_{\theta}(a_t | s_t)}{\pi(a_t | s_t)} \widehat{A}_{\pi}(s_t, a_t) \right]$$
$$\text{s.t.} \quad D_{KL}(\pi_{\theta}(\cdot \mid s) || \pi(\cdot \mid s)) \leq \delta, \quad \forall s . \tag{1}$$

In practice, we maximize this objective with a second-order approximation of the KL divergence and natural gradient descent, and replace the worst-case KL constraints over all possible states with an approximation of the mean KL based on the states observed in the current trajectory.

**Proximal policy optimization.** One disadvantage of the TRPO algorithm is that it can be computationally costly—the step direction is estimated with nonlinear conjugate gradients, which requires the computation of multiple Hessian-vector products. To address this issue, Schulman et al. (2017) propose proximal policy optimization (PPO), which tries to enforce a trust region with a different objective that does not require computing a projection. Concretely, PPO proposes replacing the KL-constrained objective (1) of TRPO by clipping the objective function directly as:

$$\max_{\theta} \mathbb{E}_{(s_t, a_t) \sim \pi} \left[ \min \left( \text{clip} \left( \rho_t, 1 - \varepsilon, 1 + \varepsilon \right) \widehat{A}_{\pi}(s_t, a_t), \ \rho_t \widehat{A}_{\pi}(s_t, a_t) \right) \right] \tag{2}$$

where

$$\rho_t = \frac{\pi_{\theta}(a_t | s_t)}{\pi(a_t | s_t)}. \tag{3}$$

Note that this objective can be optimized without an explicit projection step, leading to a simpler parameter update during training. In addition to its simplicity, PPO is intended to be faster and more sample-efficient than TRPO (Schulman et al., 2017).

**Trust regions in TRPO and PPO.** Enforcing a trust region is a core algorithmic property of different policy gradient methods. However, whether or not a trust region is enforced is not directly observable from the final rewards. So, how does this algorithmic property vary across state-of-the-art policy gradient methods?

In Figure 2 we measure the mean KL divergence between successive policies in a training run of both TRPO and PPO-M (PPO without code-level optimizations). Recall that TRPO is designed specifically to constrain this KL-based trust region, while the clipping mechanism of PPO attempts to approximate it. Indeed, we find that TRPO precisely enforces this trust region (this is unsuprising, and nearly by construction).

We thus turn our attention to the trust regions induced by training with PPO and PPO-M. First, we consider mathematically the contribution of a single state-action pair to the gradient of the PPO objective, which is given by

$$\nabla_{\theta} L_{PPO} = \begin{cases} \nabla_{\theta} L_{\theta} & \frac{\pi_{\theta}(a|s)}{\pi(a|s)} \in [1 - \epsilon, 1 + \epsilon] \text{ or } L_{\theta}^{C} < L_{\theta} \\ 0 & \text{otherwise} \end{cases},$$

$$\text{where} \quad L_{\theta} := \mathbb{E}_{(s,a) \in \tau \sim \pi} \left[ \frac{\pi_{\theta}(a|s)}{\pi(a|s)} A_{\pi}(s, a) \right],$$

$$\text{and} \quad L_{\theta}^{C} := \mathbb{E}_{(s,a) \in \tau \sim \pi} \left[ \text{clip} \left( \frac{\pi_{\theta}(a|s)}{\pi(a|s)}, 1 - \varepsilon, 1 + \varepsilon \right) A_{\pi}(s, a) \right]$$

are respectively the standard and clipped versions of the surrogate objective. As a result, since we initialize $\pi_{\theta}$ as $\pi$ (and thus the ratios start all equal to one) the first step we take is identical to a maximization step over the *unclipped* surrogate objective. It thus stands to reason that the nature of the trust region enforced is heavily dependent on the *method* with which the clipped PPO objective

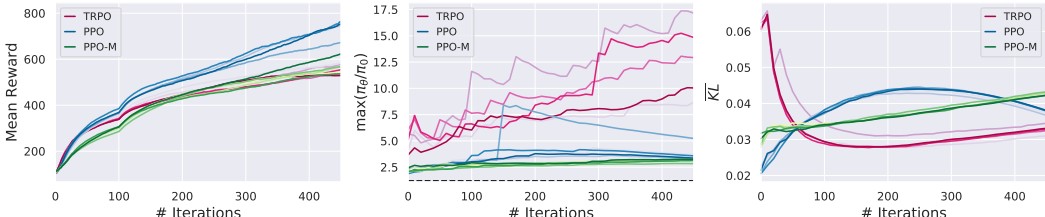

Figure 2: Per step mean reward, maximum ratio (c.f. (2)), mean KL, and mean KL for agents trained to solve the MuJoCo Humanoid-v2 task. The quantities are measured over the state-action pairs collected in the *training step*. Each line represents a training curve from a separate agent. The black dotted line represents the $1 + \epsilon$ ratio constraint in the PPO algorithm, and we measure each quantity every twenty five steps. We take mean and max KL over the KL divergences between the conditional distributions induced by the current and previous policy on the observed states in training (at each step). In the left plot we see the reward for each trained agent. From in the middle plot, we can see that the PPO variants' maximum ratios consistently violates the ratio "trust region." In the right plot, we see that both PPO and PPO-M constraint the KL well (compared to the TRPO bound of 0.07). The two methods exhibit different behavior: while PPO-M KL trends up as the number of iterations increases, PPO KL peaks halfway through training before trending down again. We measure the quantities over a *heldout* set of state-action pairs and find little qualitative difference in the results (seen in Figure 4 in the appendix), suggesting that TRPO does indeed enforce a mean KL trust region. We show plots for additional tasks in the Appendix in Figure 3. We detail our experimental setup in Appendix A.1.

is optimized, rather than on the objective itself. Therefore, the size of step we take is determined solely be the steepness of the surrogate landscape (i.e. Lipschitz constant of the optimization problem we solve), and we can end up moving arbitrarily far from the trust region. We hypothesize that this dependence of PPO on properties of the optimizer rather than the optimization objective contributes to the brittleness of the algorithm to hyperparameters such as learning rate and momentum, as observed by Henderson et al. (2018) and others.

The results we observe (shown in Figure 2) corroborate this intuition. For agents trained with optimal parameters, all three algorithms are able to maintain a KL-based trust region. First, we note that all three algorithms fail to maintain a ratio-based trust region, despite PPO and PPO-M being trained directly with a ratio-clipping objective. Furthermore, the nature of the KL trust region enforced differs between PPO and PPO-M, despite the fact that the core algorithm remains constant between the two methods; while PPO-M KL trends up as the number of iterations increases, PPO KL peaks halfway through training before trending down again.

The findings from this experiment and the corresponding calculations demonstrate that perhaps a key factor in the behavior of PPO-trained agents even from an algorithmic viewpoint comes from auxiliary optimizations, rather than the core methodology.

## 5 IDENTIFYING ROOTS OF ALGORITHMIC PROGRESS

State-of-the-art deep policy gradient methods are comprised of many interacting components. At what is generally described as their core, these methods incorporate mechanisms like trust region-enforcing steps, time-dependent value predictors, and advantage estimation methods for controlling the exploitation/exploration trade-off (Schulman et al., 2015b). However, these algorithms also incorporate many less oft-discussed optimizations (cf. Section 3) that ultimately dictate much of agent behavior (cf. Section 4). Given the need to improve on these algorithms, the fact that such optimizations are so important begs the question: *how do we identify the true roots of algorithmic progress in deep policy gradient methods?*

Unfortunately, we find that answering this question is not easy. Going back to our study of PPO and TRPO, it is widely believed (and claimed) that the key innovation of PPO responsible for its improved performance over the baseline of TRPO is the ratio clipping mechanism discussed in Section 4. However, we have already shown that this clipping mechanism is insufficient theoretically

Table 2: Full ablation of step choices (PPO or TRPO) and presence of code-level optimizations measuring agent performance on benchmark tasks. TRPO+ is a variant of TRPO that uses PPO inspired code-level optimizations, and PPO-M is a variant of PPO that does not use PPO's code-level optimizations (cf. Section 3). We find that varying the use of code-level optimizations impacts performance significantly more than varying whether the PPO or TRPO step is used. We detail our experimental setup in Appendix A.1. We train at least 80 agents for each estimate (more for some high-variance cases). We present 95% confidence intervals computed via a 1000-sample bootstrap. We also present the AAI and ACLI metrics discussed in Section 5, which attempt to quantify the relative contribution of algorithmic choice vs. use of code-level optimizations respectively.

| | MuJoCo Task | | |
| STEP | WALKER2D-v2 | HOPPER-v2 | HUMANOID-v2 |
| --- | --- | --- | --- |
| PPO | 3292 [3157, 3426] | 2513 [2391, 2632] | 806 [785, 827] |
| PPO-M | 2735 [2602, 2866] | 2142 [2008, 2279] | 674 [656, 695] |
| TRPO | 2791 [2709, 2873] | 2043 [1948, 2136] | 586 [576, 596] |
| TRPO+ | 3050 [2976, 3126] | 2466 [2381, 2549] | 1030 [979, 1083] |
| AAI | 242 | 99 | 224 |
| ACLI | 557 | 421 | 444 |

to maintain a trust region, and also that the *method* by which the objective is optimized appears to have significant effect on the resulting trust region. If code-level optimizations are thus (at least partially) responsible for algorithmic properties of PPO, is it possible that they are also a key factor in PPO's improved performance?

To address this question, we set out to further disentangle the impact of PPO's core clipping mechanism from its code-level optimizations by once again considering variations on the PPO and TRPO algorithms. Specifically, we examine how employing the core PPO and TRPO steps changes model performance while controlling for the effect of code-level optimizations identified in standard implementations of PPO (in particular, we focus on those covered in Section 3). (Note that these code-level optimizations are largely algorithm-independent: they can be straightforwardly applied or lightly adapted to any policy gradient method.) The previously introduced PPO-M algorithm corresponds to PPO *without* these optimizations. To further account for their effects, we study an additional algorithm which we denote as *TRPO+*, consisting of the core algorithmic contribution of TRPO in combination with PPO's code-level optimizations as identified in Section 3 [5]. We note that TRPO+ together with the other three algorithms introduced (PPO, PPO-M, and TRPO; all listed in Table 1) now capture all combinations of core algorithms and code-level optimizations, allowing us to study the impact of each in a fine-grained manner.

As our results show in Table 2, it turns out that code-level optimizations contribute to algorithms' increased performance often *significantly more* than the choice of algorithm (i.e., using PPO vs. TRPO). For example, on Hopper-v2, PPO and TRPO see 17% and 21% improvements (respectively) when equipped with code-level optimizations. At the same time, for all tasks after fixing the choice to use or not use optimizations, the core algorithm employed does not seem to have a significant impact on reward. In Table 2 we quantify this contrast through the following two metrics, which we denote *average algorithmic improvement* (AAI) and *average code-level improvement* (ACLI):

$$\text{AAI} = \max\{|\text{PPO} - \text{TRPO+}|, |\text{PPO-M} - \text{TRPO}|\},$$

$$\text{ACLI} = \max\{|\text{PPO} - \text{PPO-M}|, |\text{TRPO+} - \text{TRPO}|\}.$$

In short, AAI measures the maximal effect of switching step algorithms (from PPO to TRPO or vice-versa), whereas ACLI measures the maximal effect of adding in code-level optimizations for a fixed choice of step algorithm.

**PPO without clipping.** Given the relative insignificance of the step mechanism compared to the use of code-level optimizations, we are prompted to ask: *to what extent is the clipping mechanism*

---

[5]We also add a new code-level optimization, a *KL decay*, inapplicable to PPO but meant to serve as the analog of Adam learning rate annealing.

Table 3: Comparison of PPO performance to PPO without clipping. We find that there is little difference between the rewards attained between the two algorithms on the benchmark tasks. Note that both algorithms use code-level optimizations; our results indicate that the clipping mechanism is often of comparable or lesser importance to the use of code-level optimizations. We detail our experimental setup in Appendix A.1. We train at least 80 agents for each estimate (for some high-variance cases, more agents were used). We present 95% confidence intervals computed via a 1000-sample bootstrap. We also present results from the OpenAI baselines (Dhariwal et al., 2017) repository where available.

| | WALKER2D-V2 | HOPPER-V2 | HUMANOID-V2 |
|---|---|---|---|
| PPO | 3292 [3157, 3426] | 2513 [2391, 2632] | 806 [785, 827] |
| PPO (BASELINES) | 3424 | 2316 | — |
| PPO-M | 2735 [2602, 2866] | 2142 [2008, 2279] | 674 [656, 695] |
| PPO-NOCLIP | 2867 [2701, 3024] | 2371 [2316, 2424] | 831 [798, 869] |

*of PPO actually responsible for the algorithm's success?* In Table 3, we assess this by considering a PPO-NOCLIP algorithm which makes use of common code-level optimizations (by gridding over the best possible combination of such optimizations) but does not employ a clipping mechanism (this is the same algorithm we studied in Section 4 in the context of trust region enforcement)—recall that we list all the algorithms studied in Table 1.

It turns out that the clipping mechanism is not necessary to achieve high performance—we find that PPO-NOCLIP performs uniformly better than PPO-M, despite the latter employing the core PPO clipping mechanism. Our results suggest that the introduction of code-level optimizations outweighs even the core PPO algorithm in terms of effect on rewards. In fact, we find that with sufficient hyperparameter tuning, PPO-NOCLIP often matches the performance of standard PPO, which *includes* a standard configuration of code-level optimizations[6]. We also include benchmark PPO numbers from the OpenAI `baselines` repository (Dhariwal et al., 2017) where available to put results into context.

Our results suggest that it is difficult to attribute success to different aspects of policy gradient algorithms without careful analysis.

# 6 CONCLUSION

In this work, we take a first step in examining how the mechanisms powering deep policy gradient methods impact agents both in terms of achieved reward and underlying algorithmic behavior. Wanting to understand agent operation from the ground up, we take a deep dive into the operation of two of the most popular deep policy gradient methods: TRPO and PPO. In doing so, we identify a number of "code-level optimizations"—algorithm augmentations found only in algorithms' implementations or described as auxiliary details in their presentation—and find that these optimizations have a drastic effect on agent performance.

In fact, these seemingly unimportant optimizations fundamentally change algorithm operation in ways unpredicted by the conceptual policy gradient framework. Indeed, the optimizations have a profound effect on the nature of the trust region enforced by policy gradient algorithms, even controlling for the surrogate objective being optimized. We go on to test the importance of code-level optimizations in agent performance, and find that PPO's marked improvement over TRPO (and even stochastic gradient descent) can be largely attributed to these optimizations.

Overall, our results highlight the necessity of designing deep RL methods in a *modular* manner. When building algorithms, we should understand precisely how each component impacts agent training—both in terms of overall performance and underlying algorithmic behavior. It is impossible to properly attribute successes and failures in the complicated systems that make up deep RL methods without such diligence. More broadly, our findings suggest that developing an RL toolkit

---

[6]Note that it is possible that further refinement on the code-level optimizations could be added on top of PPO to perhaps improve its performance to an even greater extent (after all, PPO-NOCLIP can only express a subset the training algorithms covered by PPO, as the latter leaves the clipping severity $\varepsilon$ to be free parameter)

will require moving beyond the current benchmark-driven evaluation model to a more fine-grained understanding of deep RL methods.

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

## A   APPENDIX

### A.1   EXPERIMENTAL SETUP

All the hyperparameters used in this paper were obtained through grid searches. For PPO the exact code-level optimizations and their associated hyperparameters (e.g. coefficients for entropy regularization, reward clipping, etc.) were taken from the OpenAI baselines repository [7], and gridding is performed over the value function learning rate, the clipping constant, and the learning rate schedule. In TRPO, we grid over the same parameters (replacing learning rate schedule with the KL constraint), but omit the code-level optimizations. For PPO-NoClip, we grid over the same parameters as PPO, in addition to the configuration of code-level optimizations (since we lack a good reference for what the optimal configuration of these optimizations is). For TRPO+ we also grid over the code-level optimizations, and also implement a "KL schedule" whereby the KL constraint can change over training (analogous to the learning rate annealing optimization in PPO). Finally, for PPO-M, we grid over the same parameters as PPO (just learning rate schedules), without any code-level optimizations. The final parameters for each algorithm are given below:

Table 4: Hyperparameters for all algorithms for Walker2d-v2.

|  | PPO | TRPO | PPO-NoClip | PPO-M | TRPO+ |
|---|---|---|---|---|---|
| Timesteps per iteration | 2048 | 2048 | 2048 | 2048 | 2048 |
| Discount factor ($\gamma$) | 0.99 | 0.99 | 0.99 | 0.99 | 0.99 |
| GAE discount ($\lambda$) | 0.95 | 0.95 | 0.85 | 0.95 | 0.95 |
| Value network LR | 0.0003 | 0.0003 | 0.0006 | 0.0002 | 0.0001 |
| Value network num. epochs | 10 | 10 | 10 | 10 | 10 |
| Policy network hidden layers | [64, 64] | [64, 64] | [64, 64] | [64, 64] | [64, 64] |
| Value network hidden layers | [64, 64] | [64, 64] | [64, 64] | [64, 64] | [64, 64] |
| KL constraint ($\delta$) | N/A | 0.04 | N/A | N/A | 0.07 |
| Fisher estimation fraction | N/A | 0.1 | N/A | N/A | 0.1 |
| Conjugate gradient steps | N/A | 10 | N/A | N/A | 10 |
| Conjugate gradient damping | N/A | 0.1 | N/A | N/A | 0.1 |
| Backtracking steps | N/A | 10 | N/A | N/A | 10 |
| Policy LR (Adam) | 0.0004 | N/A | 7.25e-05 | 0.0001 | N/A |
| Policy epochs | 10 | N/A | 10 | 10 | N/A |
| PPO Clipping $\varepsilon$ | 0.2 | N/A | 1e+32 | 0.2 | N/A |
| Entropy coeff. | 0 | 0 | -0.01 | 0 | 0 |
| Reward clipping | [-10.0, 10.0] | – | [-30, 30] | – | [-10.0, 10.0] |
| Gradient clipping ($\ell_2$ norm) | -1 | -1 | 0.1 | -1 | 1 |
| Reward normalization | returns | none | rewards | none | returns |
| State clipping | [-10.0, 10.0] | – | [-30, 30] | – | [-10.0, 10.0] |

All error bars we plot are 95% confidence intervals, obtained via bootstrapped sampling.

---

[7] https://github.com/openai/baselines

Table 5: Hyperparameters for all algorithms for Humanoid-v2.

|  | PPO | TRPO | PPO-NoClip | PPO-M | TRPO+ |
|---|---|---|---|---|---|
| Timesteps per iteration | 2048 | 2048 | 2048 | 2048 | 2048 |
| Discount factor ($\gamma$) | 0.99 | 0.99 | 0.99 | 0.99 | 0.99 |
| GAE discount ($\lambda$) | 0.95 | 0.95 | 0.85 | 0.95 | 0.85 |
| Value network LR | 0.0001 | 0.0003 | 5e-05 | 0.0004 | 5e-05 |
| Value network num. epochs | 10 | 10 | 10 | 10 | 10 |
| Policy network hidden layers | [64, 64] | [64, 64] | [64, 64] | [64, 64] | [64, 64] |
| Value network hidden layers | [64, 64] | [64, 64] | [64, 64] | [64, 64] | [64, 64] |
| KL constraint ($\delta$) | N/A | 0.07 | N/A | N/A | 0.1 |
| Fisher estimation fraction | N/A | 0.1 | N/A | N/A | 0.1 |
| Conjugate gradient steps | N/A | 10 | N/A | N/A | 10 |
| Conjugate gradient damping | N/A | 0.1 | N/A | N/A | 0.1 |
| Backtracking steps | N/A | 10 | N/A | N/A | 10 |
| Policy LR (Adam) | 0.00015 | N/A | 2e-05 | 9e-05 | N/A |
| Policy epochs | 10 | N/A | 10 | 10 | N/A |
| PPO Clipping $\varepsilon$ | 0.2 | N/A | 1e+32 | 0.2 | N/A |
| Entropy coeff. | 0 | 0 | 0.005 | 0 | 0 |
| Reward clipping | [-10.0, 10.0] | – | [-10.0, 10.0] | – | [-10.0, 10.0] |
| Gradient clipping ($\ell_2$ norm) | -1 | -1 | 0.5 | -1 | 0.5 |
| Reward normalization | returns | none | returns | none | returns |
| State clipping | [-10.0, 10.0] | – | [-10.0, 10.0] | – | [-10.0, 10.0] |

Table 6: Hyperparameters for all algorithms for Hopper-v2.

|  | PPO | TRPO | PPO-NoClip | PPO-M | TRPO+ |
|---|---|---|---|---|---|
| Timesteps per iteration | 2048 | 2048 | 2048 | 2048 | 2048 |
| Discount factor ($\gamma$) | 0.99 | 0.99 | 0.99 | 0.99 | 0.99 |
| GAE discount ($\lambda$) | 0.95 | 0.95 | 0.925 | 0.95 | 0.95 |
| Value network LR | 0.00025 | 0.0002 | 0.0004 | 0.0004 | 0.0002 |
| Value network num. epochs | 10 | 10 | 10 | 10 | 10 |
| Policy network hidden layers | [64, 64] | [64, 64] | [64, 64] | [64, 64] | [64, 64] |
| Value network hidden layers | [64, 64] | [64, 64] | [64, 64] | [64, 64] | [64, 64] |
| KL constraint ($\delta$) | N/A | 0.13 | N/A | N/A | 0.04 |
| Fisher estimation fraction | N/A | 0.1 | N/A | N/A | 0.1 |
| Conjugate gradient steps | N/A | 10 | N/A | N/A | 10 |
| Conjugate gradient damping | N/A | 0.1 | N/A | N/A | 0.1 |
| Backtracking steps | N/A | 10 | N/A | N/A | 10 |
| Policy LR (Adam) | 0.0003 | N/A | 6e-05 | 0.00017 | N/A |
| Policy epochs | 10 | N/A | 10 | 10 | N/A |
| PPO Clipping $\varepsilon$ | 0.2 | N/A | 1e+32 | 0.2 | N/A |
| Entropy coeff. | 0 | 0 | -0.005 | 0 | 0 |
| Reward clipping | [-10.0, 10.0] | – | [-2.5, 2.5] | – | [-10.0, 10.0] |
| Gradient clipping ($\ell_2$ norm) | -1 | -1 | 4 | -1 | 1 |
| Reward normalization | returns | none | rewards | none | returns |
| State clipping | [-10.0, 10.0] | – | [-2.5, 2.5] | – | [-10.0, 10.0] |

## A.2 PPO CODE-LEVEL OPTIMIZATIONS

---

**Algorithm 1** PPO scaling optimization.

---

1: **procedure** INITIALIZE-SCALING()
2: $\quad R_0 \leftarrow 0$
3: $\quad RS = $ RUNNINGSTATISTICS() $\qquad\qquad$ ▷ New running stats class that tracks mean, standard deviation
4: **procedure** SCALE-OBSERVATION($r_t$) $\qquad\qquad\qquad\qquad\qquad$ ▷ Input: a reward $r_t$
5: $\quad R_t \leftarrow \gamma R_{t-1} + r_t$ $\qquad\qquad\qquad\qquad\qquad\qquad$ ▷ $\gamma$ is the reward discount
6: $\quad$ ADD($RS, R_t$)
7: $\quad$ **return** $r_t/$STANDARD-DEVIATION($RS$) $\qquad\qquad\qquad$ ▷ Returns scaled reward

---

## A.3    TRUST REGION OPTIMIZATION

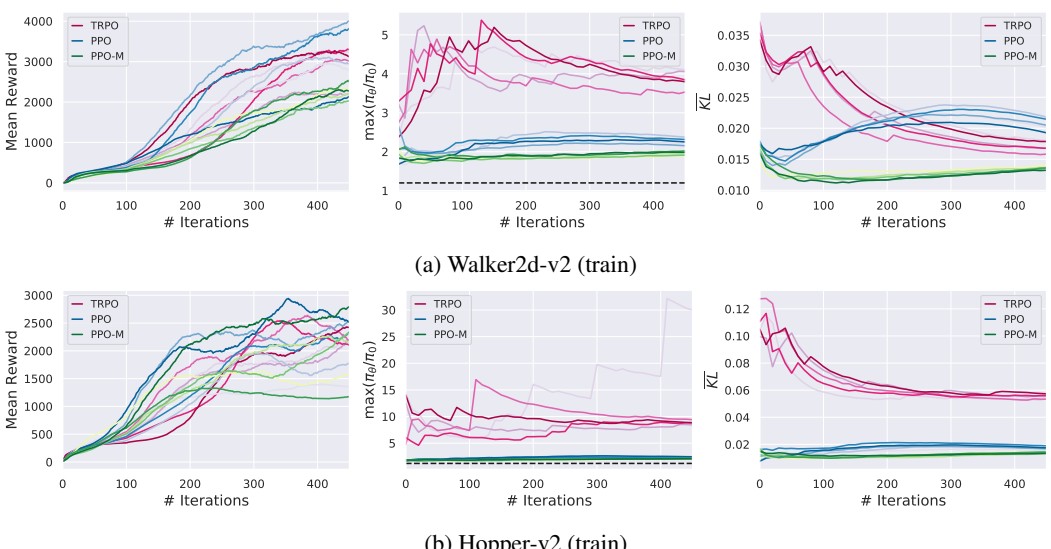

(a) Walker2d-v2 (train)

(b) Hopper-v2 (train)

Figure 3: Per step mean reward, maximum ratio (c.f. (2)), mean KL, and maximum versus mean KL for agents trained to solve the MuJoCo Humanoid task. The quantities are measured over the state-action pairs collected in the *training step*. Each line represents a training curve from a separate agent. The black dotted line represents the $1 + \epsilon$ ratio constraint in the PPO algorithm, and we measure each quantity every twenty five steps. Compare the results here with Figure 4; they are qualitatively nearly identical.

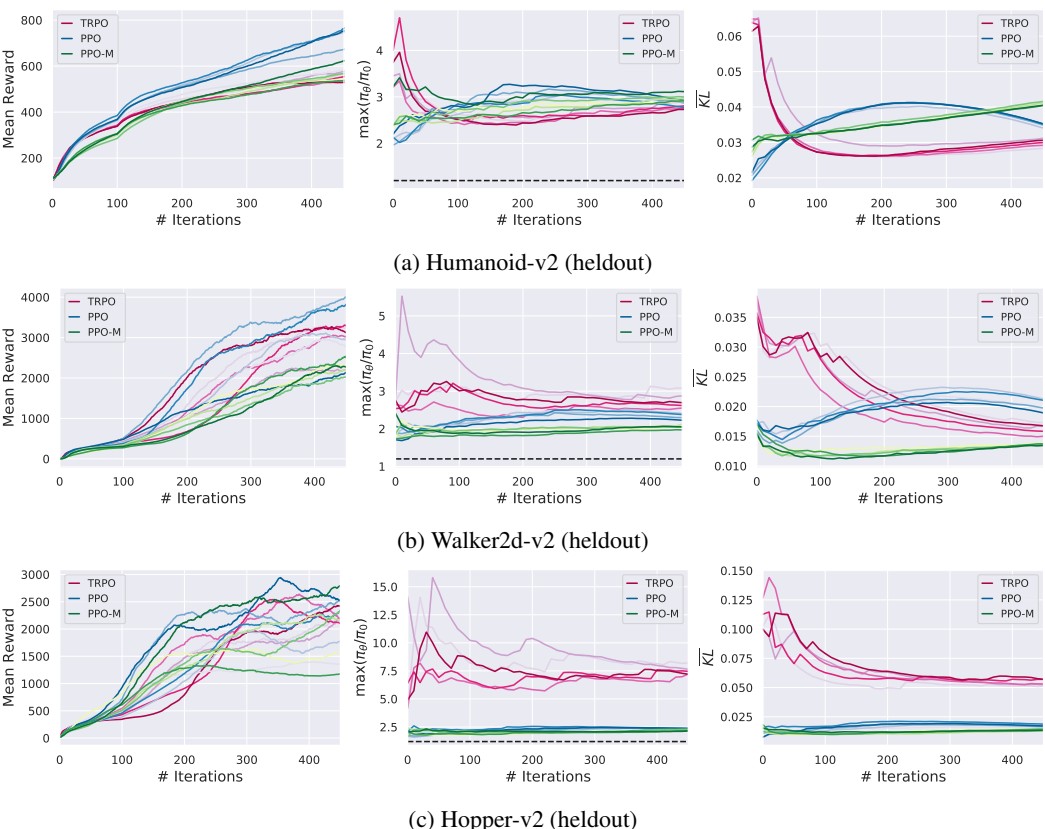

(a) Humanoid-v2 (heldout)

(b) Walker2d-v2 (heldout)

(c) Hopper-v2 (heldout)

Figure 4: Per step mean reward, maximum ratio (c.f. (2)), mean KL, and maximum versus mean KL for agents trained to solve the MuJoCo Humanoid task. The quantities are measured over state-action pairs collected from *heldout trajectories*. Each line represents a curve from a separate agent. The black dotted line represents the $1 + \epsilon$ ratio constraint in the PPO algorithm, and we measure each quantity every twenty five steps. See that the mean KL for TRPO nearly always stays within the desired mean KL trust region (at $0.06$).

