# OpenReview forum: "Implementation Matters in Deep RL: A Case Study on PPO and TRPO"
_ICLR.cc/2020/Conference — Accept (Talk)_

### Official Review · AnonReviewer1 · 2019-10-19
**Official Blind Review #1**

**Rating:** 8

**Review:**

Summary

This paper calls to attention the importance of specifying all performance altering implementation details that are current inherent in the state-of-the-art deep policy gradient community. Specifically, this paper builds very closely on the work started by  Henderson et al. 2017, building a conversation around the importance of more rigorous and careful scientific study of published algorithms. This paper identifies many "code-level optimizations" that account for the differences between the popular TRPO and PPO deep policy gradient algorithms. The paper then subselects four of these optimizations and carefully investigates their impact on the final performance of each algorithm. The clear conclusion from the paper is that the touted algorithmic improvement of PPO over TRPO has negligible effect on performance, and any previously reported differences are due only to what were considered unimportant implementation details.

Review

This paper investigates the claims made by Schulman et al. 2017 carefully, by investigating the impact of PPO's clipping mechanism on maintaining a valid trust-region; the central claim made by PPO's originating paper. The empirical results suggest that PPO is not sufficient for maintaining a valid trust-region, however the "code-level optimizations" that differ between the TRPO implementation the PPO implementation are sufficient. The ablation study of the four optimizations studied by the paper shows dramatic and clear results suggesting that annealing stepsizes and normalize rewards make very strong differences in learning performance; much more effect than demonstrated by the differences between TRPO and PPO's core algorithmic contribution as demonstrated in Figure 2 and even more strongly in Figure 3. I find the work included in this paper to be novel and a valuable contribution to the field.

For the above reasons, I recommend to accept this paper for publication at ICLR. In the following paragraphs I will discuss why I only recommend a weak accept instead of a strong accept.

My primary concern with the empirical study is the use of only three random seeds. As demonstrated in Henderson et al. 2017 (which is heavily cited in this paper), using such a small number of random seeds can have very misleading results. Although the effects appear very strong in the empirical studies in this paper, the effects likewise appear strong in Henderson et al.'s Figure 6 where 10 random seeds were split into two groups for the same algorithm. For this paper to make such strong claims about the negligence of the careful scientific study on TRPO and PPO, it would be best if this paper included far more random seeds in its investigation.

My second concern is with the discussion and conclusions drawn from Tables 1 and 2. It appears that the inclusion of clipping plays a strong role in the variance of each algorithm on every domain except Hopper. Specifically, the algorithms that include clipping appear to be much lower variance than the algorithms including clipping. Admittedly using only 3 seeds means that investigating the variance appropriately is near impossible (see the above paragraph), however variance should be considered and discussed in a conversation about the effects of the core contribution of PPO. If clipping leads to more consistent results across runs, even if those results are a little worse, it is still a valid and important contribution.

The paper cites Henderson et al. 2017 in several places. I would point out (perhaps in the introduction) that this paper builds on work already done in Henderson et al. 2017. Specifically, Henderson et al. 2017 investigates the effects of using different codebases for TRPO and shows that these different codebases result in dramatically different performance. The similarity to the investigation in this paper to too close to be unreported. However, I find that the investigation in this paper is much more complete and insightful than that of Henderson et al. 2017 (this paper has a more narrow focus), thus contributes significantly and meaningfully to this ongoing conversation.

Additional Comments (do not affect score)

It might be worthwhile to move the related work section to the beginning of the paper, either merged with the introduction or immediately after. This section is of critical importance to understanding the scope of this paper and for understanding why you are studying what you study. In fact, there is already a bit of duplication between the related works and introduction sections, so the paper could likely gain some additional real-estate by combining these.

I disagree with the terminology "code-level optimizations" and I find that it is misleading. This caused a bit of confusion on my first pass reading the paper, as I originally was expected the code differences to be more akin to using Tensorflow vs PyTorch or switching hash table functions, etc. Instead the changes focused on in this paper are changes to the problem specification and algorithm implementation. These are not simply implementation details as "code-level optimizations" suggests, but are rather details that necessarily must be included in peer-reviewed works. I don't have a suggested name to switch to, but felt strongly enough to mention it.

**Experience Assessment:**

I have published one or two papers in this area.

**Review Assessment: Checking Correctness Of Derivations And Theory:**

N/A

**Review Assessment: Checking Correctness Of Experiments:**

I carefully checked the experiments.

**Review Assessment: Thoroughness In Paper Reading:**

I read the paper thoroughly.

---

> ### Author Response · Authors · 2019-11-13
> **Response**
>
> Thank you for your detailed review and comments, which we have taken into account in our (now uploaded) revision of the manuscript. We address each point raised in the original review below:
>
> Three random seeds: For the ablation study, we only used three random seeds for computational reasons (as every random seed requires running all of the hyperparameter configurations). However, it appears as though the reviewer is (rightfully) more concerned with Tables 1 and 2---both of these, however, actually used 10 random seeds rather than 3. We have added this to their captions to clarify this issue. We will also run a few more random seeds and update the means and variances accordingly when the results become available (we cannot guarantee this before the rebuttal deadline, however.)
>
> Discussing variances in Table 2: We agree that a discussion of a possible variance reduction induced by clipping would improve the paper. In order to make sure that the apparent reduction in variance is not spurious we will be sure to run more random agents—if the trend remains, we will certainly include a discussion of how clipping might serve to reduce the variance in PPO rather than its “conventionally perceived” purpose of ensuring monotonic reward increase.
>
> Discussion of Henderson et al: We agree with the reviewer that our work builds on that of Henderson et al, and certainly did not intend to imply otherwise (hence the extensive citation noted by the reviewer). We have taken the reviewer’s advice and moved the related work to Section 2, and added the suggested mention of building on Henderson et al there.
>
> “Code-level optimizations”: We initially chose to use the term “code-level optimization” to indicate that these were algorithmic optimizations that are for the most part found only in the code of RL algorithms. However, we appreciate the reviewer’s point that the term may cause confusion. To avoid this confusion and for lack of a better term, we have, in our revision, added a footnote which indicates precisely what we mean by “code-level optimization.” We would be happy to amend this footnote or change the term if our fix has not alleviated the issue.

---

> > ### Comment · AnonReviewer1 · 2019-11-15
> > **Response**
> >
> > Thank you for addressing my concerns. I am much happier knowing that 10 random seeds were used for Tables 1 and 2. After reading other reviews and discussions on this paper, I firmly believe that this paper should be accepted.
> >
> > This, of course, does not mean that I find the paper entirely without fault. If I were to add anything to this paper, it would be to see if these results continue to hold even on smaller domains with simpler neural networks. Perhaps a domain on the scale of a simple gridworld task, where a small single layer neural network could appropriately solve the task. In this setting, dozens of random seeds could be tested overnight on a modern laptop and strong statements of statistical significance could be made. While less indicative of impact on real-world tasks, this would at least ease concerns that differences are due to chance. If the hypotheses hold true on a small gridworld with high statistical significance, and appear to hold true on a demonstration across larger tasks, then we can reasonably expect a meaningful contribution was found.

---

### Official Review · AnonReviewer3 · 2019-10-23
**Official Blind Review #3**

**Rating:** 8

**Review:**


# Summary
The papers studies the effects of code-level optimization on the performance of TRPO and PPO. Details, usually considered as implementation-level particularities, are shown to be of crucial importance for the algorithms' performance.

# Decision
The paper makes an important point, it is written clearly, and the body of evidence is convincing. Therefore, I recommend this paper for publication.

# Suggestions
Make it more clear what is meant by code-level optimizations.
    - In Sec. 2, there is a link to Appendix A.2 for a "full list", but the list in A.2 does not contain all points from Sec. 2.
    - For PPO-M, it is said "implements only the core of the algorithm". What exactly does that mean?
    - PPO-NoClip is defined as "PPO without clipping". Does it mean that it includes all other tricks apart from clipping? Please, be explicit in such places.


**Experience Assessment:**

I have published one or two papers in this area.

**Review Assessment: Checking Correctness Of Derivations And Theory:**

I assessed the sensibility of the derivations and theory.

**Review Assessment: Checking Correctness Of Experiments:**

I assessed the sensibility of the experiments.

**Review Assessment: Thoroughness In Paper Reading:**

I made a quick assessment of this paper.

---

> ### Author Response · Authors · 2019-11-13
> **Response**
>
> Thank you for your review and comments on our paper. We have made sure to more precisely define code-level optimizations (algorithmic changes that are predominantly found in codebases and not presented as core parts of their respective RL algorithms, see our reply to R1 for more detail), PPO-M (PPO but without any of the code-level optimizations mentioned in Sec 2 or the Appendix), and PPO-NoClip (PPO minus the clipping, including all the optimizations). We have also copied the optimizations listed in Section 2 to the appendix, so that the appendix contains a complete list of the optimizations.

---

> > ### Comment · AnonReviewer3 · 2019-11-13
> > **Clarification of suggestions**
> >
> > Thanks for your reply.
> >
> > What I mean by "being more clear about what is meant by code-level optimization" is not that you provide a definition but that you explicitly write which optimizations are included and which are not included directly at the point where you introduce PPO-M, PPO-NoClip, etc. Going back and forth between Sect. 2 and Appendix is inconvenient, and I was especially confused that the list of "code-level" optimizations in Sect. 2 is different from the one in Appendix; therefore, it was not clear to me what exactly is included and what not.

---

> > > ### Author Response · Authors · 2019-11-13
> > > **Updated revision**
> > >
> > > We have moved all the optimizations listed to the main text, added a table summarizing the different algorithms, and have written clarifications in the sections describing each new algorithm we present. Thank you for your feedback, and let us know if there is any more clarification needed!

---

### Official Review · AnonReviewer2 · 2019-10-23
**Official Blind Review #2**

**Rating:** 8

**Review:**

This paper investigates the impact of implementation "details", with existing implementations of TRPO and PPO as examples. The main takeaway is that the performance gains observed in PPO (compared to TRPO) are actually caused by differences in implementation, and not by the differences between the two learning algorithms. In particular, adding to the TRPO code the same implementation changes as in PPO makes TRPO on par with (and possibly even better than) PPO. The clipping objective of PPO is also found to have no significant impact on its performance. This calls for more careful comparisons between algorithms (by minimizing implementation changes and more in-depth ablation studies) than has typically been done until now in the RL research community.

Although this paper is pretty straightforward and does not bring meaningful algorithmic improvements, I still believe it should be accepted as reproducibility and evaluation are a major issue in RL, and people need to be aware of these kinds of implementation differences that can affect the reported results.

My only important concern is that I could not find a link to the code, which I believe is a must for such a paper focusing on implementation. Could the authors please confirm that they will release their code?

Other small remarks:
	- Fig. 1 is hard to read, I think more synthetic results could have easily conveyed more clearly the intended message
	- When referring to Fig. 2 and 3 please specify "left", "middle" or "right"
	- Fig. 2's caption should describe the plots in left to right order (also what does "maximum versus mean KL" mean?)
	- Fig. 3's caption lists mean KL twice on its first line
	- "The trust region for PPO-NoClip bounds KL to a lesser degree": this is confusing as it sounds like it is "less bounded" while it is actually "more bounded" (as said in Fig. 3's caption)
	- It would help comparing Fig. 2 and Fig. 3 if they both used the same y axis range
	- Typo: "enforcing" => enforces

Update after author feedback: increasing score to "Accept" thanks to the release of the code

**Experience Assessment:**

I have read many papers in this area.

**Review Assessment: Checking Correctness Of Derivations And Theory:**

N/A

**Review Assessment: Checking Correctness Of Experiments:**

I carefully checked the experiments.

**Review Assessment: Thoroughness In Paper Reading:**

I read the paper thoroughly.

---

> ### Author Response · Authors · 2019-11-13
> **Response**
>
> Thank you for your comments on our paper. With regards to the reviewer’s main concern, we completely agree and are definitely planning to release code for this work along with the final version. We have been working on making the code more readable, modular, and easy to run, and will include a GitHub link with the final version of the paper. (We are almost done with cleaning up the codebase, if we are done before the revision deadline we will upload an anonymous copy and link it, but either way a link will appear in the final version.)
>
> We address the other minor comments below:
> - We experimented with many different plot styles and visualization techniques for Figure 1 and converged on this version due to readability and its ability to express the relatively intricate data collected. However, we have updated the caption of Figure 1 to better describe the plot style (as it is somewhat unconventional), hopefully alleviating this concern.
> - We have updated both the captions of Figure 2 and 3 to fix the noted issues, and also added (left), (middle), and (right) to our references to Figures 2 and 3
> - We have fixed various typos/confusing wordings, including those found by the reviewer. Thank you for pointing them out!

---

> > ### Comment · AnonReviewer2 · 2019-11-13
> > **Re: Response**
> >
> > Thank you for addressing my concerns & remarks.

---

> > > ### Author Response · Authors · 2019-11-15
> > > **Re: Re: Response**
> > >
> > > We are glad to hear that our response addressed your concerns!
> > >
> > > Since you brought up the issue of code explicitly in your review, we wanted to point out our latest revision and general comment, where we posted code for reproducing our results, with toggles for all of the relevant code-level optimizations discussed in our work.

---

### Public Comment · ~Erik_Wijmans1 · 2019-11-07
**Effect of Normalized Advantage?**

Hi,

I really like this paper, the implementation-level additions in TRPO and PPO have always been confusing as to whether or not they really matter (and it seems like they do really matter!).  One addition that has always confused me is normalized advantage (in baselines: https://github.com/openai/baselines/blob/master/baselines/ppo2/model.py#L139 and in this popular pytorch implementation: https://github.com/ikostrikov/pytorch-a2c-ppo-acktr-gail/blob/master/a2c_ppo_acktr/algo/ppo.py#L36), but I don't see it as one of the things that you investigated.  Is this something you considered?  If not, I would be very curious to know if that one matters!

---

> ### Author Response · Authors · 2019-11-09
> **Thanks!**
>
> Thanks for the kind comment! We actually studied all algorithms with normalized advantage in place (in a sense we treated normalization as part of the advantage estimation process, rather than as part of the RL algorithm itself). Studying the effect of normalization would be interesting future investigation! (note that even TRPO/other RL algorithms use normalized advantage as well.)

---

### Author Response · Authors · 2019-11-15
**Code Release**

Dear Reviewers,

We finished our cleaning and documentation of the codebase before the rebuttal deadline! As such, we have revised our paper to include a link to the (anonymized) codebase, which contains extendable, modular, and commented implementations of PPO and TRPO, with precise control over all the code-level optimizations. We also provide utilities for reproducing the results of this work.

The code can be found here: https://github.com/implementation-matters/code-for-paper.

We will continue to improve the code over time to make it even easier to use, but we believe it is definitely in a good enough state (and is sufficiently important) to be added to the paper now.

---

### Author Response · Authors · 2020-04-13
**Camera ready and revision changes**

We have now uploaded the camera-ready. While preparing to upload, we confirmed a bug in our code (precisely, in computing KL-divergences), and thus reran all of our experiments to ensure their validity. The vast majority of the results went unchanged, with the exception of the explicit plots of KL divergence for the PPO-NoClip algorithm, which have thus been removed (along with the respective 2-3 sentences in the text.) We also made a few improvements to the paper for clarity. A full summary is below:
- Added numbers from OpenAI baselines to compare to
- More thorough explanation of gridding procedure and hyperparamters used
- Increased sample size for both the ablation study (5 agents per config for a total of 320 agents trained) as well as the rewards tables (now > 80 agents per cell, with 95% bootstrapped confidence intervals)
- Fix typos/editing for clarity

---

### Public Comment · ~Zelin_Zhao1 · 2021-04-05
**One issue about value clipping.**

In your paper, you state that L^V = min(clipped value reward, unclipped reward), but in your implementation (https://github.com/MadryLab/implementation-matters/blob/5ee6ecb12545365d9178135e65576adfc0d82f52/src/policy_gradients/steps.py#L96)  and in OpenAI standard implementation, you use L^V = max(clipped value reward, unclipped reward). Is this a problem?

---

> ### Author Response · Authors · 2021-04-05
> **Re: One issue about value clipping**
>
> Thanks for the comment! This is actually just a typo in the paper---the equation should just say max instead of min. As you mentioned, the standard OpenAI implementation also uses max.

---

### Decision · Program_Chairs · 2019-12-19

**Decision:**

Accept (Talk)

**Comment:**

This paper provides a careful and well-executed evaluation of the code-level details of two leading policy search algorithms, which are typically considered implementation details and therefore often unstated or brushed aside in papers. These are revealed to have major implications for the performance of both algorithms.

The reviewers are all in agreement that this paper has important reproducibility and evaluation implications for the field, and adds substantially to our body of knowledge on policy gradient algorithms. I therefore recommend it be accepted.

However, a serious limitation is that only 3 random seeds were used to get average performance in the first, key experiment. Experiments are expensive, but that result is not meaningful without more runs, and arguably could be misleading rather than informative. The authors should increase the number of runs as much as possible, at least to 10 but ideally more.